# Challenges and Opportunities in the Clinical Development of STING Agonists for Cancer Immunotherapy

**DOI:** 10.3390/jcm9103323

**Published:** 2020-10-16

**Authors:** Leila Motedayen Aval, James E. Pease, Rohini Sharma, David J. Pinato

**Affiliations:** 1Department of Surgery & Cancer, Imperial College London, Hammersmith Hospital, Du Cane Road, London W120HS, UK; lma16@ic.ac.uk (L.M.A.); r.sharma@imperial.ac.uk (R.S.); 2Inflammation, Repair & Development, National Heart & Lung Institute, Imperial College London, London SW7 2AZ, UK; j.pease@imperial.ac.uk

**Keywords:** cGAS, STING, immune checkpoint inhibitors (ICI), cyclic dinucleotides (CDNs), small molecule agonist, STING agonists, immune therapy, drug delivery systems (DDS)

## Abstract

Immune checkpoint inhibitors (ICI) have revolutionised cancer therapy. However, they have been effective in only a small subset of patients and a principal mechanism underlying immune-refractoriness is a ‘cold’ tumour microenvironment, that is, lack of a T-cell-rich, spontaneously inflamed phenotype. As such, there is a demand to develop strategies to transform the tumour milieu of non-responsive patients to one supporting T-cell-based inflammation. The cyclic guanosine monophosphate-adenosine monophosphate synthase-stimulator of interferon genes (cGAS-STING) pathway is a fundamental regulator of innate immune sensing of cancer, with potential to enhance tumour rejection through the induction of a pro-inflammatory response dominated by Type I interferons. Recognition of these positive immune-modulatory properties has rapidly elevated the STING pathway as a putative target for immunotherapy, leading to a myriad of preclinical and clinical studies assessing natural and synthetic cyclic dinucleotides and non-nucleotidyl STING agonists. Despite pre-clinical evidence of efficacy, clinical translation has resulted into disappointingly modest efficacy. Poor pharmacokinetic and physiochemical properties of cyclic dinucleotides are key barriers to the development of STING agonists, most of which require intra-tumoral dosing. Development of systemically administered non-nucleotidyl STING agonists, or conjugation with liposomes, polymers and hydrogels may overcome pharmacokinetic limitations and improve drug delivery. In this review, we summarise the body of evidence supporting a synergistic role of STING agonists with currently approved ICI therapies and discuss whether, despite the numerous obstacles encountered to date, the clinical development of STING agonist as novel anti-cancer therapeutics may still hold the promise of broadening the reach of cancer immunotherapy.

## 1. Introduction

Novel cancer immunotherapies such as anti-cytotoxic T-lymphocyte-associated protein-4 (CTLA-4) and anti-programmed death-1/programmed death-ligand-1 (PD-1/PD-L1) monoclonal antibodies have revolutionised cancer therapy [1,2]. Immune checkpoint inhibitors (ICI) can re-establish immune surveillance by targeting negative regulatory checkpoint molecules in tumour cells to bypass their immune evasion strategies of diminishing T-cell reactivity and inducing immune exhaustion [3,4]. Despite success in the clinic, response rates to these therapies varies greatly among patients. For example, ipilimumab yields a response rate of only ~15% and rarely exceeds 25% for those treated with anti-PD-1/PD-L1 ICI [5,6,7]. The efficacy of ICI appears to correlate with a presence of a pre-existing pro-inflammatory tumour microenvironment (TME) evidenced by higher immune cell infiltration or PD-L1 expression, whereas “immune desert” or “cold” tumours appear to respond less to ICI (TME) [8,9]. In order to observe the maximal effects of ICI-therapy, ‘cold’ tumours must be reprogrammed towards an immunogenic and pro-inflammatory or ‘hot’ phenotype which can reinvigorate anti-tumour immunity [10,11]. A promising approach involves upregulation of the cyclic guanosine monophosphate-adenosine monophosphate synthase-stimulator of interferon (IFN) genes (cGAS-STING) pathway, a major component of the innate immune system involved in antiviral and anti-tumour immunity [12]. Profound therapeutic effects of drugs targeting this pathway have been demonstrated in multiple preclinical murine tumour models and there are multiple ongoing clinical trials assessing the efficacy of STING agonists as monotherapies or in combination with ICI [13]. However, these drugs pose profound challenges for in vivo application, specifically their physiochemical properties, which render them poorly membrane permeable and prone to rapid enzymatic degradation [14,15,16]. This article provides an updated review on the latest progress of the development of cyclic dinucleotide (CDN) and non-nucleotidyl small-molecule STING agonists and their preclinical and clinical trials as cancer immunotherapies. Meanwhile, there is emphasis on the limitations of these therapies and the future directions required to overcome them, notably the development of novel drug delivery systems (DDS). 

## 2. The cGAS-STING Pathway

The cGAS-STING pathway (Figure 1) is an innate immune pathway that senses cytosolic double stranded (ds)DNA and results in a host of downstream signalling events in response to infection [17]. The pathway bridges innate and adaptive immunity and promotes dendritic cell (DC) priming, migration and subsequent cytotoxic T-lymphocyte (CTL) priming and cytotoxicity at the tumour site. Hence, the pathway has the ability to overcome the immunosuppressive environment of some cancers and could sensitise of patients to ICI-therapy [17,18,19,20]. Firstly, the enzyme cGAS binds to the sugar-phosphate backbone of cytosolic dsDNA in a sequence-independent manner [21]. Upon binding, conformational changes in cGAS allow the catalysis of ATP and GTP into cyclic Gp(2′,5′)Ap(3′,5′) (cGAMP), an endogenous mammalian CDN [22,23,24]. The adaptor protein STING is then activated by cGAMP at the endoplasmic reticulum which leads to its oligomerisation and tetramerization after which STING translocates to the Golgi [25,26]. STING can also be directly activated by bacteria-derived CDNs, such as cyclic di-GMP [27]. At the Golgi, STING recruits and activates TANK binding kinase-1 which then phosphorylates the transcription factors interferon regulatory factor 3 (IRF3) and nuclear factor kappa-light-chain-enhancer of activated B cells [28,29,30]. IRF3 translocates to the nucleus where it exerts its function in upregulating the expression of immune stimulated genes (ISGs) and Type I IFNs which results in the release of cytokines such as IFN-ß and the Th1 cell recruiting chemokines [21,24]. Subsequently, there is activation and migration of immune cells including DCs, T-cells and natural killer (NK) cells to the target cell [31]. 

Type I IFNs are essential to the cross-priming of tumour-specific T-cells by tumour infiltrating antigen presenting cells (APCs). Indeed, mice lacking STING or IRF3, but not other innate signalling pathways, failed to achieve natural CD8^+^ anti-tumour T-cell priming [12,31,32,33,34]. Self-DNA does not ordinarily activate the pathway as it is packaged within cell nuclei, however if exposed to the cytoplasm it can trigger an autoimmune response [7,35]. To prevent chronic activation of STING and toxic concentrations of IFN, there are multiple autoregulatory mechanisms in place including lysosomal degradation of STING, degradation of cGAMP by ecto-nucleotide pyrophosphatase/phosphodiesterase-1 (ENPP1) and activation of the pathway itself induces autophagy, resulting in clearance of cytosolic DNA [15,36,37]. The STING signalling pathway can be activated in a host of immune cells including T-cells, DCs, macrophages, B-cells and also in NK cells, which are then primed for cytotoxic killing of cancer cells [38]. 

**Figure 1 jcm-09-03323-f001:**
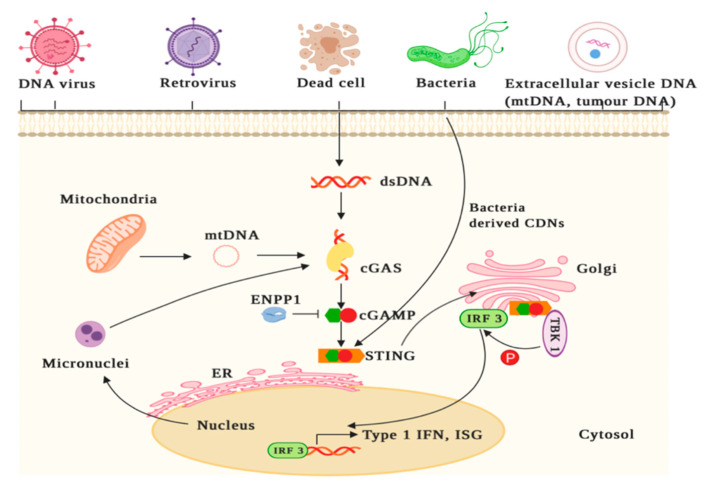
The cyclic guanosine monophosphate-adenosine monophosphate synthase-stimulator of interferon genes (cGAS-STING) signalling pathway. cGAS binds to the sugar-phosphate backbone of cytosolic double stranded (ds)DNA of a viral, apoptotic, exosomal, mitochondrial, or micronuclei nature, in a sequence-independent manner [21]. This binding results in conformational changes to cGAS which facilitates the catalysis of ATP and GTP into cyclic Gp(2′,5′)Ap(3′,5′) (cGAMP) [22,23,24]. As an autoregulatory mechanism, cGAMP is degraded by ecto-nucleotide pyrophosphatase/phosphodiesterase 1 (ENPP1) [15,36,37]. STING is then activated by cGAMP at the endoplasmic reticulum (ER), resulting in its oligomerisation and tetramerization after which it translocates to the Golgi [25,26]. STING can also be directly activated by bacteria derived cyclic dinucleotides (CDNs) [27]. STING then recruits and activates TANK binding kinase 1 (TBK1) which phosphorylates the transcription factor interferon regulatory factor 3 (IRF3) [28,29]. Phosphorylated IRF3 then translocates to the nucleus and upregulates the expression of Type I interferons (IFNs) which results in the release of pro-inflammatory cytokines such as IFN-β [21,24]. This is an original figure created with BioRender.com.

## 3. cGAS-STING and Cancer Immunity

The cGAS-STING pathway is now also recognised as a crucial mechanism regulating tumour immunity (Figure 2) [39]. The activation of this pathway may prevent early neoplastic progression through upregulation of ISGs as well as by mediating the secretion of cytokines, chemokines, and proteases belonging to the senescence-associated secretory phenotype, which collectively restrict tumorigenesis [40,41,42]. Cancer cells harbour unstable genomes which are prone to chromosomal mis-segregation during cell division and generate micronuclei [43]. These micronuclei are susceptible to rupture and release genomic contents into the cytosol which is then detected by cGAS [44,45]. Mitochondria also play a role in the activation of the cGAS-STING pathway in cancer as malignant cells release mitochondrial DNA into the cytosol under oxidative stress and mitochondrial malfunction (Figure 1) [46]. 

Antigen-specific priming of T-cells can occur via a mechanism involving the transfer of tumour-cell-derived cGAMP into immune cells which then activates the cGAS-STING pathway (Figure 2) [47,48]. Cancer cells do not typically produce Type I IFNs, but through the mechanisms discussed above, constitutively active cGAS is an established characteristic of many tumour cells, and as a result, cGAMP is also produced. cGAMP-producing tumour cells can activate the STING pathway in nearby host cells through cGAMP acting as an immunotransmitter [48,49]. This CDN enters cells through the folate transporter, SLC19A1, and subsequently activates the STING pathway in host immune cells [47]. Transfer of cGAMP from cancer cells into NK cells results in the production of cytokines which enhance NK cell cytotoxicity to potentiate full or partial tumour rejection [48]. In immature DCs, uptake of cGAMP also results in the production of pro-inflammatory cytokines which facilitate DC activation and maturation [48]. Furthermore, DCs mature through the recognition, endocytosis and presentation of tumour-associated antigens which are exposed as a result of cancer cell death [50]. DCs present peptide fragments via their MHCI or MHCII molecules to CD8^+^ and CD4^+^ T-cells, respectively, in the lymph node [50]. This results in the priming and activation of tumour specific CTLs which infiltrate the tumour milieu where they recognise, bind and lyse tumour cells, releasing more tumour antigens and amplifying the cancer-immunity cycle [50]. 

The importance of cGAS-STING signalling in cancer is supported by preclinical models of tumours with lost or reduced STING expression [46,51,52]. For example, loss of cGAS-STING signalling in hepatocellular carcinoma resulted in enhanced tumorigenicity and decreased CTL infiltration in non-small-cell lung carcinoma (NSCLC) [46,52]. However, cGAS-knockout mice were not prone to spontaneous tumour development, supporting the fact that various tumour-suppressing pathways are able to prevent tumorigenesis [41]. Additionally, only a subset of patients across malignancies can generate a natural anti-tumour T-cell response due to cGAS-STING activation, whilst others will show no such spontaneous immunity, underscoring the complexity of the role of this pathway in tumorigenesis [53]. Nonetheless, this pathway is critical in cancer immunity and the presence of primed CD8^+^ TILs has been reported to be of even more powerful prognostic value than classical TNM tumour staging [54,55,56,57].

**Figure 2 jcm-09-03323-f002:**
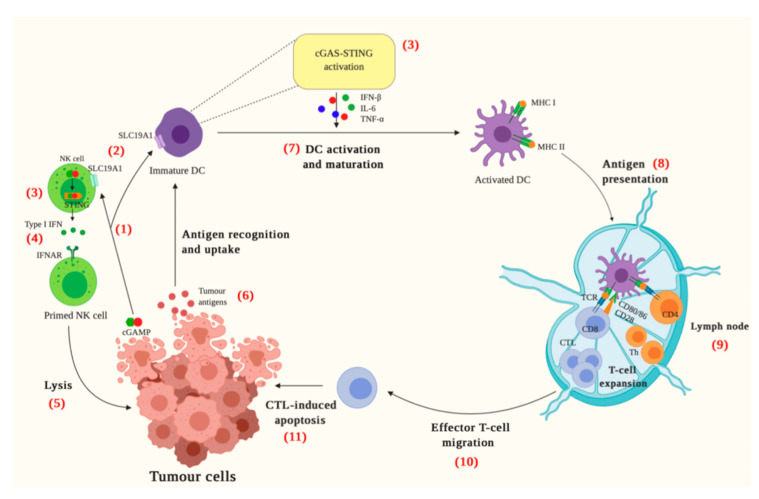
A schematic diagram of the role of the cGAS-STING pathway in cancer immunity. cGAMP can be recognised by host immune cells, notably DCs and NK cells (1) [47,48,50]. cGAS is constitutively active in most tumour cells and results in the production of cGAMP which can be internalised by NK cells and immature DCs through the folate transporter SLC19A1 (2). This results in the activation of cGAS-STING in these cells (3) [47]. In NK cells, the STING-dependent production of Type I IFNs (4) enhances NK cell cytotoxicity and subsequent tumour cell lysis (5) [47,48,49]. Immature DCs can be activated and mature through the recognition, uptake and processing of tumour antigens (6) as well as by pro-inflammatory cytokines produced as a result of cGAMP uptake and subsequent cGAS-STING activation (7) [47,48,50]. The mature/activated DCs display antigens via MHC molecules (8) and present them to T cells in the lymph node, potentiating T cell differentiation into CTLs or Th cells (9). The primed CTLs the migrate to and infiltrate the tumour site (10) resulting in cancer cell death and the release of more tumour antigens, perpetuating the cancer-immunity cycle (11) [50]. This is an original figure created with BioRender.com.

## 4. STING Activating Drugs

Recent years have seen a rapidly growing interest in the development of synthetic and natural CDN analogues as well as non-nucleotidyl small molecule STING agonists. Most STING agonists are natural CDNs derived from bacterial or human sources and function by mimicking the native ligand of STING, 2′3′-cGAMP [58]. c-di-GMP, c-di-AMP and 3′3′-cGAMP have been discovered in prokaryotic cells (Figure 3) [30]. STING is found intracellularly within the ER meaning STING agonists must penetrate the cell membrane to exert their effects [13]. However, natural CDNs are electronegative, hydrophilic and highly susceptible to enzymatic degradation by phospho-diesterases, primarily ENPP1 [36,37,59]. These characteristics alongside their large size render them impermeable to cell membranes, lead to low drug bioavailability in tumour tissues, and narrow their therapeutic windows [13]. To address these limitations, various approaches are being taken such as the development of CDN DDS, synthetic CDNs, small molecule STING agonists, and ENPP1 inhibitors. Indeed, the first-in-class ENPP1 inhibitor MAVU-104, developed by Mavupharma, is orally active and expected to enter clinical trials later in 2020 [60]. 

Originally, 5,6-dimethylxanthenone-4-acetic acid (DMXAA), a flavonoid with putative anti-tumour activity provided a proof-of-concept for the use of STING agonists to strengthen tumour immunity [20,61,62,63,64,65,66,67,68].

### 4.1. DMXAA

Initially developed as a neo-vasculature disrupting agent, the small molecule STING agonist DMXAA, failed to improve outcome of patients with NSCLC when co-administered with standard-of-care chemotherapy in a Phase III clinical trial despite promising results seen in preclinical murine models [67]. Despite mouse and human STING (mSTING and hSTING, respectively) sharing 68% amino acid identity, DMXAA has specificity for mSTING but not hSTING [63,69]. Further studies on hSTING revealed that due to STING polymorphisms, DMXAA is unable to bind to hSTING [61,62,63,64]. However, whether the anti-angiogenic effects of DMXAA are STING dependent is still unclear, and therefore, the reason for failure in the Phase III trial may be due to more than its inability to bind hSTING. These proof-of-concept studies allowed DMXAA to provide a prototypic structural model for further development of both natural and synthetic STING agonists. However, one of the existing challenges in evaluating the mechanism of action of DMXAA and other CDNs is whether their predominant mechanism of action relates to direct antiproliferative effect mediated by a Type-1 interferon response as opposed to an active role in remodelling the TME and suppress immune tolerance.

### 4.2. Natural STING Agonists—Natural CDNs 

The anti-tumour potential of these compounds was first demonstrated using c-di-GMP which successfully inhibited basal proliferation of human colon carcinoma cells in vitro [70]. It was also later revealed that CDNs are potent inducers of a host Type I IFN response in bone marrow macrophages via a STING-dependent mechanism [71,72,73]. Due to this property, agonists of STING such as c-di-GMP and c-di-AMP have been successfully used as cancer vaccine adjuvants and 2′3′-cGAMP was found to improve responses to radiation-based cancer therapy [74,75,76]. Injection of 3′3′-cGAMP into mice with either chronic lymphocytic leukaemia or multiple myeloma resulted in apoptosis and tumour regression, implicating the ability of STING agonists to directly eradicate malignant B cells alongside their immunostimulatory functions [77]. In fact, 3′3′-cGAMP was shown to activate STING more efficiently than DMXAA, in assays of STING, IRF3 and STAT1 phosphorylation [77]. Disappointingly, 3′3′-cGAMP has a higher binding affinity for mSTING than hSTING, making it of limited use in humans [61]. Further, 2′3′-cGAMP is the only known naturally occurring mammalian CDN and has distinct characteristics from bacterial CDNs, notably its remarkably high affinity for STING (Figure 3) and greater efficacy in inducing Type I IFNs [30]. Intratumoral (i.t.) injection of 2′3′-cGAMP in the CT26 mouse colon adenocarcinoma model reduced tumour size and improved survival and when injected in 4T1-luc (breast cancer), B16F10 (melanoma), mSCC1 (murine squamous cell carcinoma) murine models, it also delayed tumour growth [78]. In another study, injection with 2′3′-cGAMP in the B16F10 model yielded similar observations as well as simultaneous reduction of lung metastases [33]. 

### 4.3. Synthetic CDNs

Li et al. synthesised a 2′3′-cGAMP analogue, 2′3′-cG^S^A^S^MP, which was resistant to ENPP1 hydrolysis and showed similar affinity for hSTING in vitro as well as ten-fold more potent induction of IFN-β secretion from THP-1 cells. This bisphosphothioate analogue was also ~40 times more resistant to ENPP1 hydrolysis than the naturally occurring ligand, rendering it a desirable compound for future development [15,79]. Gajewski et al. synthesised various synthetic CDN-derivatives and selected those which displayed binding affinity for all hSTING alleles while retaining the ability to bind mSTING [20]. The compounds they developed, namely ML RR-S2 CDG, ML RR-S2 cGAMP, and ML RR-S2 CDA (ADU-S100), bear dithio mixed-linkages with both R,R- and R,S-dithio diastereomers (Figure 4). The authors found that the compounds were resistant to digestion with phosphodiesterases and induced greater expression of Type I IFNs in THP-1 cells when compared to endogenous CDNs. ADU-S100 was the only compound selected for clinical advancement and showed improved stability, lipophilicity, and significantly enhanced STING activation when compared with cGAMP and other prokaryotic derived CDNs. I.t. injection resulted in profound tumour regression in B16 melanoma, CT26 colon, and 4T1 breast cancer murine models, the anti-tumour effects of which were shown to be long lasting. Importantly, unlike ML RR-S2 CDG, mice receiving ADU-S100 showed higher overall survival (OS). Additionally, when re-challenged with the same tumour cell line, the mice failed to develop tumours and growth of distal, untreated tumours of a different cancer type were also suppressed. Mice bearing CT26 tumours showed no protection to tumour re-challenge by a different tumour type, 4T1, indicating that the immune memory established is tumour specific [20]. ADU-S100 is currently undergoing a Phase II trial in combination with pembrolizumab in first-line head and neck squamous cell carcinoma (HNSCC) (Section 6.1) (NCT03937141). Recently, Eisai Inc. developed a novel class of macrocycle-bridged STING agonists (MBSAs) which showed a high degree of chemical and metabolic stability due to the conformational rigidity of the macrocycle bridge [80,81]. E7766, an MBSA derivative of ADU-S100, demonstrated pan-genotypic activity across seven hSTING genotypes in human primary cells and long-lived anti-tumour activity in murine tumour models with no serious adverse events [80,81]. In March 2020, Eisai Inc. initiated a Phase Ia/Ib clinical trial of E7766 (Section 6.8) (NCT04144140). Other than ADU-S100, further synthetic CDN compounds have been developed and are undergoing clinical trials [13]. The statuses of these compounds will be discussed in Section 6. 

### 4.4. Additional Non-Nucleotidyl Small Molecule STING Agonists 

Presently, there are eleven examples of non-CDN agonists (Table 1), a majority of which share structural similarities with DMXAA (Figure 5) [82,83,84,85,86,87,88,89,90,91,92,93,94]. GlaxoSmithKline (GSK) identified small molecule STING agonists, which shared a vital amidobenzimidazole component [91]. They found that ABZI binds to the C-terminal domain of STING with two molecules bound per each STING dimer, prompting them to create a single dimeric ligand (diABZI) to improve binding affinity [91]. diABZI had an increased binding affinity by more than 1000-fold and they confirmed that the linker molecule did not interact with the STING protein. The compound induced dose-dependent STING activation in PBMCs with a 400-fold greater potency than cGAMP and displayed remarkable selectivity for STING (Table 1). Their potential as systemically administered therapeutics was evaluated in a model of colorectal tumours (CT-26) in BALB/c mice. Treatment resulted in significant tumour regression and significantly improved survival with 80% of mice remaining tumour free after trial cessation [91]. However, it should be highlighted that the potent therapeutic effects of diABZI in treating this tumour were measured under the experimental conditions that the drug was injected only two days after tumour engraftment when no tumour or TMEs were formed yet [91]. The most promising non-nucleotidyl small molecule STING agonist is diABZI, the first intravenous (i.v.) efficacious non-CDN STING agonist with systemic anti-tumour activity in murine models. Based on these findings, GSK are undergoing a Phase I clinical trial looking at a diABZI-like molecule GSK3745417 (Section 6.5) (NCT03843359). 

Curadev Pharma revealed three patents for nine bicyclic benzamides with potent hSTING agonist properties [88,89,90]. All nine compounds induced the secretion of pro-inflammatory cytokines in human PBMCs and in BALB/c mice bearing CT26 hSTING expressing tumours (Table 1). I.t. injections of the compounds resulted in profound tumour regression [88,89,90], with one of the compounds almost completely inhibiting tumour growth [89]. All compounds tested by the group displayed efficient binding to the five major polymorphic variants of the hSTING protein. Merck Sharp and Dohme Corporation recently patented a series of multi-substituted benzothiophenes as activators of hSTING [92]. The compounds were modified to include a 4-oxobutanoic acid side chain, which improved activity, as well as minor adaptations on the phenyl group which profoundly influenced their potency. Some compounds induced a level of IFN-β secretion which was several folds higher than 2′3′-cGAMP (Table 1) [92]. Two further patents from Merck evidenced the ability of these compounds to have significant STING-dependent anti-tumour activity in vivo in an MC38 murine tumour model [92]. On the back of this, a clinical study using these compounds either alone or in combination with anti-PD-1 antibodies to treat patients with recurrent and/or metastatic solid tumours or lymphomas has been proposed [95]. 

Most recently, Merck discovered an orally available non-nucleotide-based STING agonist, MSA-2 (benzothiophene oxobutanoic acid), through the use of a phenotypic screen of 2.4 million compounds for chemical inducers of IFN-β secretion [93]. The researchers demonstrated that MSA-2, when dosed either orally or subcutaneously in the MC38 syngeneic mouse tumour model, achieved dose-dependent anti-tumour activity. Impressively, complete tumour regressions were seen in 80–100% of the treated mice, the drug was well-tolerated, and a significant increase in IFN-β, IL-6 and tumour necrosis factor-α were observed in tumour and plasma. When re-challenged with the same tumour type, tumours failed to grow in 95% of the mice, indicating that MSA-2 achieved durable anti-tumour immunity [93]. Notably, the combination of MSA-2 with anti-PD-1 yielded synergistic effects in reducing tumour growth and prolonged survival compared to monotherapy, in four difficult-to-treat murine tumour types (CT26, MC38, B16F10 and LL-2) [93]. MSA-2 preferentially targets tumour tissue due to its unique mechanism of action. Experimental analyses revealed that MSA-2 displays higher cellular potency in the acidified TME compared to normal tissue due to enhanced cellular entry and retention, which most likely contributes to the detected tumour specificity and tolerability [93]. Another systemically administered non-nucleotide small molecule STING agonist, SR-717, was discovered by Chin et al. using a cGAS-STING pathway targeted cell-based screen of 100,000 druglike small molecules [94]. Crystallographgy of the compound revealed that it is a direct mimetic of STING’s naturally occurring ligand cGAMP and induced the same closed conformation of STING [94]. This property was also observed in MSA-2 [93]. Of significance, the effects of SR-717 were not influenced by interallelic or interspecies differences in amino acid sequence of STING, which was a potential reason for the clinical inefficacy of DMXAA [62,94]. Intraperitoneal administration of the compound into the poorly immunogenic and highly aggressive B16F10 tumour models on day 11 following sufficient tumour establishment; displayed maximal tumour growth inhibition, prolonged overall survival and importantly, was well-tolerated [94]. Intraperitoneal administration of SR-717 demonstrated greater efficacy compared to anti-PD-1 or anti-PD-L1 monotherapy with respect to tumour growth and overall survival. The compound also successfully prevented the formation of pulmonary nodules in a mouse melanoma metastasis model. SR-717 enhanced the activation and cross-priming of CD8+ T lymphocytes within the tumours and the draining lymph nodes (dLNs) and activated both NK and DCs within the dLNs [94]. Markedly, effacious doses of diABZI were found to induce ~20ng/mL of IFN-β, in contrast to the >0.2ng/mL induced by SR-717 which was also found to be well tolerated and efficacious [91,94]. This suggests that the anti-tumour effects of STING agonists can be achieved without the toxicity associated with possible “cytokine storm”. 

## 5. Combination Therapy—STING Agonists with ICI

Pharmacologically upregulating the STING pathway alone as an immunotherapy is unlikely to be successful as the recruited tumour specific T-cells can initiate immune inhibitory pathways including PD-L1 or FOXP3, thus preventing tumour clearance [96,97]. Alternatively, STING agonists combined with ICI could yield synergistic effects. As aforementioned, only a fraction of patients respond to ICI and even when they do, a proportion go on to relapse with a fatal, drug-resistant form of the initial disease [98,99,100]. Supporting the potential of drug-induced STING activation to sensitise patients to ICI, ICI-therapy has shown weaker therapeutic effects in STING deficient mice [12]. The presence of effector T-cells within the TME is closely associated with a favourable response to ICI-therapy [8,101,102,103]. Stimulation of STING results in the production of the crucial Th1 recruiting cytokines, CXCL9 and CXCL10 and Type I IFNs [21]. Type I IFNs promote DC activation and maturation which results in improved antigen presentation to CD4^+^ T-lymphocytes and importantly, antigen cross-presentation to CD8^+^ T-lymphocytes hence T-cell priming (Figure 2) [104]. Indeed, STING agonist therapy has been combined with ICI in multiple difficult to treat tumour models and promising results have been achieved [33,105,106,107].

Given the significant role the cGAS-STING pathway plays in cancer outcomes, there are currently a number of ongoing clinical trials assessing the anti-tumour potential of STING agonists as monotherapies or in combination with ICI. 

## 6. STING Agonists in Clinical Trials 

To date, a number of STING agonists have entered clinical testing in early-phase clinical trials. Table 2 provides an overview of the clinical candidates in current development.

### 6.1. ADU-S100 (ML-RR-S2-CDA) 

ADU-S100 is the earliest STING agonist to enter human clinical trials as a cancer immunotherapy. Details of its structure and outcomes in preclinical studies are outlined in Section 4.3. In 2015, Aduro and Novartis entered into an agreement for the research and development of ADU-S100 resulting in ADU-S100 entering Phase I clinical trials in 2016 for the treatment of advanced/metastatic solid tumours or lymphoma. This was either as a monotherapy or in combination with ipilimumab or the anti-PD-1 spartalizumab. Partial results released from the spartalizumab trial demonstrated that ADU-S100 in combination with the ICI achieved anti-tumour activity without any dose-limiting toxicities (DLTs) in 3/8 anti-PD-1 naïve triple negative breast cancer (TNBC) patients and 2/25 melanoma patients previously treated with ICI [108]. Overall, 12/53 participants in the combination group achieved stable disease (SD), 4/53 a partial response (PR) and only 1 showed complete response (CR). Those who demonstrated CR achieved a median reduction of 73% in target tumour diameter. Unfortunately, pharmacokinetic study showed rapid absorption of ADU-S100 from the i.t. injection site and the terminal half-life was also short, ranging from 10–23 min. Of the reported treatment related adverse events (TRAEs), 12.2% were serious (grade 3/4) and included increased lipase levels, diarrhoea and worsened liver function [108]. Recently, Novartis and Aduro set up a Phase II trial of ADU-S100 in combination with pembrolizumab for the treatment of PD-LI positive, recurrent or metastatic HNSCC patients. However, in December 2019, Novartis pulled out of all ADU-S100 clinical trials based on dissatisfactory clinical efficacy. Nonetheless, Aduro are still continuing the Phase II study of ADU-S100 in combination with pembrolizumab for HNSCC and are also preparing to instigate a clinical trial of ADU-S100 monotherapy to treat non-muscle invasive bladder cancer (NMIBC) [109]. 

### 6.2. MK-1454

MK-1454 is a synthetic CDN analogue developed by Merck & Co for the treatment of advanced/metastatic solid tumours or lymphomas. The drug candidate entered Phase I clinical trials both as a monotherapy and in combination with pembrolizumab. Merck announced that the combination therapy arm resulted in robust and durable responses which had lasted for at least 6 months, with 24% of patients achieving PR compared to 0% (0/20) in the monotherapy arm and only 20% of patients receiving MK-1454 alone achieved SD [110]. Additionally, in the combination arm, tumour regression was seen in both injected and non-injected tumours with an 83% median reduction in size of the primary lesion. Dose escalations ranged from 10–3000 μg in the monotherapy group and 90–1500 μg for combination therapy. TRAEs were 82.6% and 82.1% in both the monotherapy and combination arms, respectively and severe TRAEs leadings to trial cessation were recorded in 7.1% (*n* = 2) of patients on combination therapy. DLTs, namely severe vomiting and injection site reactions, were reported at 1500 μg. The maximum tolerated dose (MTD) has not yet been determined and the dose escalation study is still ongoing with complete results expected in 2021. The structure of MK-1454 is undisclosed.

### 6.3. MK-2118

In 2017, Merck & Co initiated a Phase I clinical trial of another CDN STING agonist MK-2118 for the treatment of solid tumours or lymphomas. They administered the drug i.t. as a monotherapy or in combination with pembrolizumab and via subcutaneous injection with pembrolizumab. The purpose of the study was to assess the safety, tolerability, MTD of the drug as well as a recommended Phase II dose (RP2D). Final results are expected in April 2022 and the structure of MK-2118 is undisclosed. 

### 6.4. BMS-986301

The synthetic STING agonist BMS-986301 was acquired by Bristol-Myers Squibb from IFM Therapeutics in 2017 [111]. Preclinical results displayed 90% complete tumour regression in CT26 and MC38 mouse tumour models in addition to 80% complete regression in the CT26 model in combination with anti-PD-1 [112]. The ICI alone resulted in no complete regressions with the ICI alone. Identical dosing with ADU-S100 resulted in only 13% complete regression in both tumour models. All CT26 mice displaying CR also showed immunological memory, demonstrated by rejection of fresh tumour cells [112]. Given the preclinical success, the drug entered Phase I clinical trials in 2019 as a monotherapy and in combination with nivolumab or ipilimumab for the treatment of advanced solid cancers. The study is expected to be completed in 2023. The structure of BMS-986301 is undisclosed. 

### 6.5. GSK-3745417

GSK-3745417 (GlaxoSmithKline) is a synthetic non-CDN STING agonist with a di-ABZI scaffold, however its exact chemical structure is undisclosed [13]. Unlike other STING agonists, it has shown to be successful when administered i.v. In 2019, GlaxoSmithKline launched a Phase I first time in humans trial of the drug to intravenously treat 300 participants with advanced solid cancers and to assess the drug’s safety, tolerability, clinical efficacy and recommended dose both as a monotherapy and in combination with pembrolizumab. Final results are expected in 2024.

### 6.6. SB-11285

SB-11285 (Spring Bank Pharmaceuticals) is a small molecule CDN STING agonist developed as a cancer therapy. In preclinical studies, the drug induced IFN-ß with a 200-fold increased potency compared to cGAMP. In vivo, the drug displayed long-lasting and potent anti-tumour responses in A20 and CT26 murine cancer models when administered i.t., i.v., intraperitoneally (i.p.) or intramuscularly (i.m.) [113]. Subsequently, a Phase Ia/Ib clinical trial was initiated in 2019 in patients with advanced solid tumours. Phase Ia of the trial is a dose-escalation study with i.v. administration of the drug and Phase Ib involves i.v. administration of the drug in combination with nivolumab in tumour types expected to be responsive to immunotherapy. In the first quarter of 2020, the company announced a collaboration with Roche to also include their anti-PD-L1 inhibitor atezolizumab in their combination therapy group [114]. By the end of 2020, the company expects to have a RP2D to enable advancement of the clinical trial. The estimated completion date is May 2022. The structure of SB-11285 is undisclosed.

### 6.7. IMSA-101

IMSA-101 (ImmuneSensor Therapeutics) is a small-molecule cGAMP analogue undergoing an open label dose-escalation (Phase I) and dose expansion (Phase IIa) study which was designed in 2019 to evaluate the safety and clinical efficacy of IMSA-101 alone or in combination with an ICI to treat solid, refractory malignancies. The results from the Phase I study will determine the RP2D. Trial completion is expected in 2023. The structure of the STING agonist is undisclosed.

### 6.8. E7766

E7766 (Eisai Inc.) is an MBSA derivative of ADU-S100 developed which very recently entered Phase Ia/Ib clinical trials to assess the clinical efficacy, safety and tolerability as a monotherapy. It will be administered i.t. to treat advanced solid tumours and lymphomas. Results are expected in December 2022. The drug is also expected to enter a Phase I study for the treatment of NMIBC, but it is yet to start recruitment (NCT04109092).

## 7. Challenges to STING Activating Drug Development 

STING activation can be a double-edged sword when manipulated as a cancer immunotherapy. Five main issues are discussed here. The first issue is that STING activation could in fact induce a tolerogenic immune response. Lemos and colleagues showed that DNA sensed by myeloid DCs induced Indoleamine-2,3-dioxygenase (IDO) production via the cGAS-STING pathway, which subsequently suppressed effector and helper T-lymphocytes. Regulatory T-cells were also activated to promote a dominant regulatory phenotype [87]. The same group went on to show that STING-induced IDO production in the TME supported the growth of Lewis lung carcinoma and that STING ablation increased the numbers of CD8^+^ TILs and tumour cytotoxicity [96,115]. Significantly, they found that the tolerogenic response was only present in tumours bearing low antigenicity. Their findings suggest that STING agonists may not be effective in all tumour environments, and that tumour neoantigens could be used as future biomarkers to stratify patients into STING-agonist responders and non-responders [96,116]. The second issue was raised by Larkin et al. who performed a preclinical study that showed STING agonists could induce stress and death in T-lymphocytes. Their findings emphasise the caution which must be taken as STING agonists are developed and that the long-term effects of these drugs on T-cell function represents a gap in our knowledge [117]. The third issue involves the status of single nucleotide polymorphisms (SNPs) in hSTING with implications for the selection of appropriate STING agonists. Analysis of SNPs from the 1000 Genome Project revealed five main haplotypes of hSTING; WT, REF, HAQ, AQ and Q. While all haplotypes are able to recognise metazoan CDNs to varying degrees, the distribution of each haplotype is geographically distinct [118]. For example, STING AQ and Q isoforms occur mainly in the African population whereas the HAQ variant is predominantly found in Asian and Hispanic populations, which suggests there are differences within each population in the ability to respond to bacterial CDNs [118]. The fourth issue involves defining the safe dosing of STING agonists. Injection of ‘free’ small molecule STING agonists can lead to rapid dissemination in the bloodstream resulting in uncontrolled inflammation and ‘cytokine storm’, tissue toxicity, autoimmunity and a promotion of tumour growth [119]. Chronic activation of STING results in the persistent generation of cytokines, creating an inflammatory TME which can encourage tumour development [59,120]. It is essential that a therapeutic window is determined which will allow STING agonists to exert their anti-tumour effects whilst minimizing immunotoxicity. The fifth issue involves the route of administration of STING agonists as most agonists currently in clinical trials are largely focussed on i.t. delivery, which poses two main problems. Firstly, i.t. administration limits the use of such drugs to patients with accessible, solid tumours [91]. Secondly, the anti-tumour immunity induced by i.t. injection would not cover the host’s entire tumour antigen spectrum as even within one individual, tumour heterogeneity exists in distal metastases [121]. Innovative approaches are needed to unlock the tremendous potential of STING agonists. One such approach includes the development of biomaterial-based STING agonist delivery systems and results in preclinical tumour models have thus far been promising [14,122,123,124].

## 8. Future Directions 

DDS have gained much attention for the treatment of cancer. Advantages of DDS include their ability to profoundly alter and improve drug pharmacokinetics, provide targeted drug delivery, and therefore alleviate TRAEs whilst enhancing therapeutic outcomes [125,126,127,128]. The three main DDS discussed here are liposomes, polymers and hydrogels (Table 3) [14,123,124,128,129,130,131,132,133,134,135,136,137]. Liposomes are cationic lipid structures with an aqueous core, making them great candidates for encapsulating negatively charged, hydrophilic CDN compounds [138,139,140]. They can also electrostatically interact and fuse with the negatively charged cell membrane and thus enable drug delivery to intracellular STING [124,128,129,130,131]. Polymeric NPs are a suitable nanocarrier for STING agonists given their favourable properties including hydrolytic degradability in vivo, controlled drug loading and release kinetics, and overall safety [141,142]. Hydrogels are highly hydrophilic polymer networks, which facilitate local and controlled drug release, leading to the recruitment of tumour toxic immune cells [143,144]. In this section, we will discuss pertinent STING-activating DDS for cancer immunotherapies. 

### 8.1. Liposomes

Liposomes have previously been used to successfully deliver c-di-GMP to the dLNs as vaccine adjuvants [131]. The Harashima group established YSK05, a synthetic cationic lipid with high fusogenicity and formed a YSK05/c-di-GMP liposome. In vitro, they documented a large, STING-dependent induction of IFN-ß compared with the free form of c-di-GMP. In vivo, YSK05/c-di-GMP induced CTL activity and inhibited T-cell lymphoma growth [130]. I.v. administration of YSK05/c-di-GMP activated NK cells and induced the production of Type I IFNs, resulting in a profound anti-tumour effect in a lung metastasis mouse model [129]. Mooney et al. similarly designed a series of cationic liposomes with varying surface polyethylene glycol (PEG) levels and used them to encapsulate 2′3′-cGAMP [128]. Administration via both i.t. and i.v. routes induced greater 2′3′cGAMP uptake into the cytosol and higher IFN- ß gene expression as well as potent anti-tumour activity against metastatic melanoma in the lung, compared to no or reduced effect with 2′3′-cGAMP. Intratumoural administration of the formulation improved 2′3′-cGAMP retention at the tumour site as well as the localisation of 2′3′-cGAMP with tumour-specific APCs. Furthermore, PEGylated 2′3′-cGAMP induced adaptive immunity, evidenced by resistance to rechallenge by the same tumour cells in mice [128]. More recently, the Ting group used 3′3′-cGAMP-loaded cationic liposomal NPs and showed that they could drive the production of proinflammatory cytokines and nitric oxygen species which promoted macrophage repolarisation from an M2 to an M1 phenotype whilst enhancing STING-dependent anti-tumour immunity in various difficult to treat tumour models [124]. Additionally, a single i.v. dose of these NPs was sufficient to suppress tumour growth [124]. Evidently, liposomes represent a promising DDS to enhance STING agonist cancer immunotherapy. (Table 3).

### 8.2. Polymers

A biodegradable poly(beta-amino ester) (PBAE) cationic polymer was developed by Wilson et al. and was combined with ADU-S100 to form PBAE/CDN polymeric NPs [123]. They showed that PBAE/CDN NPs improved STING-dependent IRF3 production in vitro at >100-fold lower doses than the free form of CDN. When administered in vivo in combination with an anti-PD-1 ICI into murine models bearing poorly immunogenic B16-F10 melanomas, the CDN NPs showed an order of magnitude reduction in the dose required to eradicate the established tumours [123]. Another group recently described STING-activating NPs (STING-NPs), polymersomes designed for enhanced cytosolic delivery of 2′3′-cGAMP [14]. At physiologic pH, these polymersomes remained intact but after intracellular uptake, they disassembled in response to lysosomal acidification allowing escape of cGAMP into the cytosol. STING-NPs are one of the most potent DDS for STING activation and can amplify the potency of cGAMP by up to three orders of magnitude in multiple cell types and can induce a change in the TME to a ‘hot’, pro-inflammatory one. This finding was consistent in both murine and human metastatic melanoma samples, supporting the potential for clinical translation of STING-NPs. They also significantly improved responses to anti-PD-1 and anti-CTLA-4 ICI, inhibited tumour growth, induced immunological memory, increased OS and could be systemically administered [14]. The same group used this technology to successfully enhance STING activation in neuroblastoma cells as well as within the TME and showed synergistic effects when administered with anti-PD-L1 ICI [132]. The Ting group used acetalated dextran (Ace-DEX) to develop polymeric microparticles for cGAMP loading using electrospray, resulting in >90% encapsulation of cGAMP into Ace-DEX microparticles [133]. The therapeutic efficacy of cGAMP-loaded Ace-DEX was displayed using two TNBC murine models and a B16-F10 model. Four routes of administration of the loaded microparticles were able to generate robust anti-tumour immune responses, demonstrated by an up to 50 times more potent Type-1 IFN response compared to clinically used immune-activating drugs such as imiquimod. These results were achieved at a dose of 0.1ug of cGAMP, amounting to a 100-fold dose reduction compared to both PEGylated cGAMP and YSK05/c-di-GMP liposomes discussed in Section 8.1, and a 1000-fold dose reduction compared to free cGAMP [128,129,133,145,146,147]. Evidently, a significantly reduced amount of drug is required with this type of DDS, which is desirable for clinical translation as high doses of STING agonists have been shown to cause T cell apoptosis [117,148,149]. (Table 3).

### 8.3. Hydrogels

One group loaded cGAMP into linear polyethyleneimine (LPEI)/hyaluronic acid (HA) as a vaccine adjuvant [134]. HA has many advantages including its ability to adopt multiple forms and an ability to release drugs in a spatiotemporal manner [150]. The formulated hydrogel could effectively be delivered into phagocytic macrophages, hence resulting in profound induction of the cytokines IFN-β and IL-6 compared with conventional cationic liposomes [134]. In another example, the Goldberg group designed a hydrogel scaffold via cross-linking HA in a mould and then loaded it with 2′3′-cGAMP [135]. They confirmed that relative to i.v. or i.t. administration of the soluble STING, slow release of the drugs from a biodegradable hydrogel at the tumour resection site cured a higher percentage of mice with local breast cancer and reduced metastases. This was due to the therapy enhancing the number of NK cells, DCs, T cells and inducing large amounts of IFN-β. However, this intervention was deemed appropriate only for post-surgical tumour resection, as it stimulates the innate immune system during the post-operative immunosuppression associated with wound healing and so would be inappropriate to treat intact primary tumours [135]. Similarly, Gough and colleagues combined CDA with Matrigel and applied it to the resection site [136]. They chose Matrigel due to its hydrophilicity, its ability to form an amorphous solid gel when heated to body temperature and its rapid degradation in vivo. They tested their formulation in a murine model of HNSCC and found it was able to stimulate IFN-ß production and subsequent CD8^+^ T-lymphocyte expansion, which minimised tumour recurrence. As with LPEI/HA, this localised therapy is suitable in tumours undergoing surgical resection [136]. The most promising hydrogel DDS is STINGel, which was packaged with ADU-S100 and used in a challenging model of murine oral cancer [137]. STINGel was formed using a cationic multidomain peptide, which could self-assemble in to antiparallel β-sheet nanofibres and easily be delivered by syringe. Their study demonstrated successful rejection of MOC2-E6E7 tumours in wild type mice with only a single injection of STINGel, likely due to the high concentration of CDN achieved at the tumour site, which induced high immune cell recruitment and cytotoxicity. Impressively, 60% of mice exhibited a complete anti-tumour response and 100% of surviving mice exhibited no tumour growth following secondary inoculation, demonstrating immunological memory [137]. (Table 3).

## 9. Concluding Remarks

Preclinical studies have demonstrated the crucial role of STING in anticancer immunity and as a result, a plethora of STING agonists have been and are being developed. Despite the huge investment by Novartis in ADU-S100, the preliminary data from the clinical trial showed that only 1/53 patients achieved a CR in the combination group (Table 2). Similarly, Merck’s trial data has been underwhelming; a 0% overall response rate was achieved with MK-1454 monotherapy and only 24% when combined with pembrolizumab (Table 2). Neither of the studies displayed consistent observation of abscopal activity, that is, shrinkage of distal tumours, when the STING agonists were used as single agents [151]. Following on from these data, it must be determined why these therapies have only shown efficacy in some patients, what the resistance mechanisms are, and as a result, useful biomarkers could be identified which anticipate the therapeutic outcomes of STING agonist therapy. Alternatively, the varied response could be accounted for by STING SNPs, which suggests the need for personalised STING therapy. These questions underscore the need for a deeper, mechanistic understanding of STING in order to design more widely applicable, clinically effective and safe therapies. 

ADU-S100 was administered intratumorally and displayed poor pharmacokinetic properties with a short half-life. Concerningly, 6/8 of STING agonists in ongoing clinical trials are also administered i.t. However, there are promising alternative avenues such as the systemically administered ENPP1 inhibitor MAVU-104 and non-nucleotidyl small molecule STING agonists. Of these, the orally administered compound MSA-2 holds the most promise. Oral administration circumvents the need for intratumoral injection, the compound preferentially targets tumour tissue and provides a convenient, and low-cost delivery route [93]. 

This review discussed 11 non-CDN STING agonists, of which only one has entered clinical trials (GSK-3745417) and Merck’s Benzothiophene derivatives are expected to enter trials shortly [95]. Importantly, both these candidates can be administered systemically. Although non-CDN agonists hold great promise as cancer immunotherapies, the profound dangers of potential ‘cytokine storm’ still remain. To assess the true risk of systemic administration, future studies should compare the clinical efficacy, safety and tolerability of systemic administration compared to i.t. administration. As substantiated in this review, perhaps the most encouraging method to achieve efficient and safe delivery of these drugs for cancer patients is through the utilisation of DDS. The biomaterials discussed here allow colocalization, improved bioavailability and altered release kinetics of immunotherapies, thereby rendering the drugs more clinically efficacious whilst reducing off-target toxicity and potentiating systemic administration. Most DDS studied preclinically can be applied systemically thus facilitating the use of STING agonists for a wider range of patients and allowing STING activation at multiple metastatic sites (Table 3). STING-DDS in preclinical studies, alongside inducing anti-tumour immunity, enhanced drug delivery to tumour infiltrating macrophages and DCs, which are crucial in formulating a response to STING agonists [14,78,124]. Of the three DDS discussed here, polymers seem to be the most favourable candidate to be pursued in future clinical trials given their synergy with ICI therapies, which was not assessed in any of the other types (Table 3) [14,123,133]. In particular, STING-NPs impressively amplified the potency of cGAMP by up to three orders of magnitude [14]. Despite the undeniable potential, which DDS hold in improving STING agonists’ clinical efficacy, no STING-DDS have entered clinical trials which is suggested here to be the next step in determining the true clinical translatability of STING agonists. 

There have been many advances in the study of STING and its role in the immune-oncology field yet there are still many questions which need to be answered and challenges overcome. For example, a pressing issue is whether STING is even a validated immune target given the epigenetic silencing of STING in some tumour cells and the severe autoimmune diseases associated with STING overstimulation [119,152,153,154,155]. However, if successful, STING agonists could become effective and indispensable pharmacological agents for the treatment of cancer. In addition to cancer, the cGAS-STING pathway holds a promising role for the treatment of autoimmune or inflammatory diseases [156]. The multifaceted role of STING further emphasises the importance of manipulating this pathway for the treatment of various diseases and a substantial increase in the number of candidate STING agonists is to be expected in the future.

## Figures and Tables

**Figure 3 jcm-09-03323-f003:**
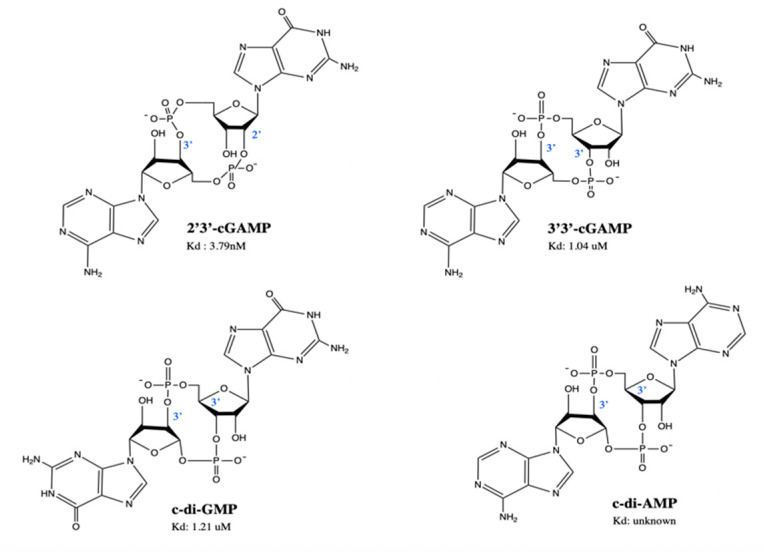
Structures of naturally occurring cyclic dinucleotides. 2′3′-cGAMP is the native ligand of STING, found in eukaryotic cells. 3′3′-cGAMP, c-di-GMP and c-di-AMP are found in prokaryotic cells. Kd (dissociation constant) represents the affinity of STING binding to the cyclic dinucleotides. This is an original figure created with ChemDraw.

**Figure 4 jcm-09-03323-f004:**
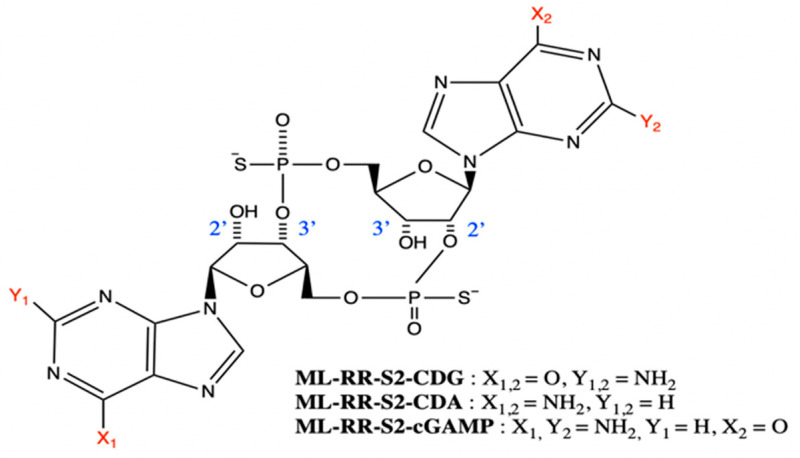
Structures of synthetic cyclic dinucleotide derivatives developed by Gajewski et al. [20]. Substituents X1, X2, Y1, and Y2 are shown inset. An original figure created with ChemDraw.

**Figure 5 jcm-09-03323-f005:**
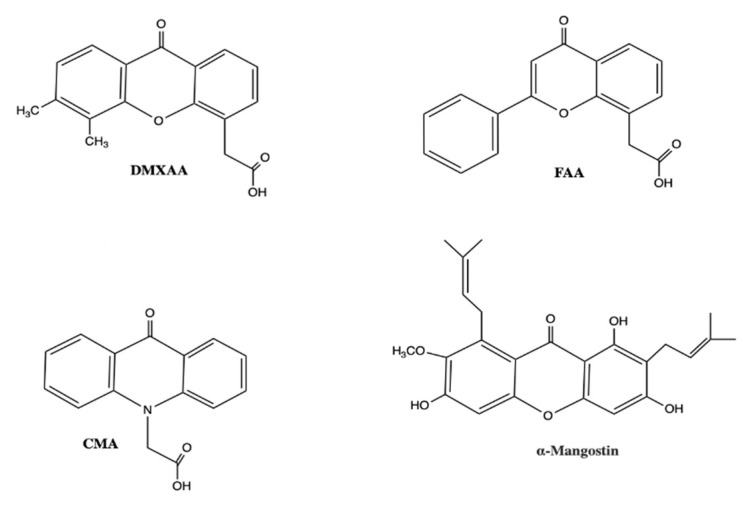
Small molecule STING agonists bearing structural similarity to DMXAA. Abbreviations: DMXAA, 5,6-dimethylxanthenone-4-acetic acid; FAA, flavone acetic acid; CMA, carboxymethyl-9-acridanone. This is an original figure created with ChemDraw.

**Table 1 jcm-09-03323-t001:** Non-nucleotidyl small molecule stimulator of interferon genes (STING) agonists and their inhibitory activities at mouse and human STING.

Small Molecule STING Agonist	hSTING Activity	mSTING Activity	Assay Used	Pertinent Findings	References
DMXAA	No	Yes	ITC	mSTING: K_d_~130 nM. hSTING K_d_: undetectable. ITC upper bound of detection ~100 uM therefore hSTING affinity for DMXAA >1000-fold lower than for mSTING.	[62]
FAA	No	Yes	Vesicular stomatitis viral inhibition assay	Murine splenic leukocytes generated 250 units/mL * of IFN following 3 h incubation with 0.25 mg/mL FAA compared with <5 units/mL produced in human peripheral blood leukocytes.	[82]
CMA	No	Yes	ELISA	CMA in murine model: strong induction of Type-1 IFN production (~1.2 pg/mL after 18 h). CMA in human cells (PBMCs, fibroblasts): failed to induce detectable cytokine responses even at 4000 ug/mL.	[83]
α-Mangostin	Yes	Yes	q-RT-PCR + IFN-ß-luciferase reporter	THP1 cells treated with 25uM α-Mangostin for 9 h significantly increased IFN-ß mRNA expression ~8-fold. HEK 293 T cells were transiently transfected with hSTING or mSTING and then transfected with up to 25 uM a-Mangostin. After 24 h, IFN- ß luciferase activity was reported in both mSTING and hSTING but was ~5-fold greater in hSTING.	[84]
BNBC	Yes	No	q-RT-PCR	BNBC concentration-dependently induced IFN-ß in HepG2/STING (reconstituted cell line with hSTING) cells but not HepG2/mSTING (reconstituted cell line with mSTING) cells. 200 uM of BNBC significantly induced IFN- ß mRNA expression ~5000-fold in HepG2/STING cells compared with only ~2-fold in HepG2/mSTING cells.	[85]
DSDP	Yes	No	q-RT-PCR	DSDP concentration-dependently induced IFN-ß in HepG2/STING cells but not HepG2/mSTING cells. 50uM of DSDP significantly induced IFN-ß mRNA expression ~300-fold in HepG2/STING cells compared with only ~1-fold with no significant difference compared to the internal control.	[86]
diABZI	Yes	Yes	IFN- ß secretion assay	In human PBMCs, diABZI induced dose-dependent activation of STING and secretion of IFNβ with an EC_50_ of 130 nM. This is more than 400-fold more potent than cGAMP. No information on mSTING activity. diABZI activated secretion of Type I IFNs and pro-inflammatory cytokines in wild type but not mice deficient in STING.	[91]
Bicyclic benzamides	Yes	No	Luciferase assay	All compounds have a micromolar range of activity in the HEK293T-hSTING luciferase assay, and potently induce the secretion of IFN- β, IL-6, TNF-αand CXCL10 in PBMCs and in BALB/c mice bearing CT26 hSTING expressing tumours.	[88,89,90]
Benzothiophenes	Yes	No	3H-cGAMP filtration binding assay	Of the 5 Benzothiophene derivatives developed by Merck Sharp and Dohme Corporation, 3 compounds show significant functional activity with percent activation (% effect) several folds higher than 2′3′-cGAMP in IFN- ß secretion of THP1 cells. Compound 1: EC_50_: 9335nmol/L, % effect = 290% Compound 2: EC_50_:2575nmol/L, % effect = 298% Compound 3: EC_50_:4073nmol/L, % effect = 394% % effect values represent the ability to induce IFN-β secretion in THP-1 cells relative to 2’,3’-cGAMP at 30 μmol/L.	[92]
MSA-2	Yes	Yes	AlphaLISA + competitive radioligand binding assay	EC_50_ of 8.3 and 24 μM for human STING isoforms WT and HAQ, respectively. MSA-2 shows antitumor activity and stimulates interferon-β secretion in tumours, induces tumour regression with durable antitumor immunity, and synergizes with anti-PD-1 in the LL-2 tumour model. It exhibits dose-dependent antitumor activity when administered by IT, SC, or PO routes, and dosing regimens were identified that induced complete tumour regressions in 80 to 100% of treated animals. MSA-2 (PO: 60 mg/kg or SC: 50 mg/kg; single dose) effectively inhibited tumour growth induced substantial elevations of IFN-β, interleukin-6 (IL-6), and TNF-α in MC38 mouse tumour model. Stepwise reductions of extracellular pH from 7.5 to 6 increased MSA-2 potency in both THP-1 cells and mouse macrophages, potency of cGAMP was unchanged with pH changes.	[93]
SR-717	Yes	Yes	q-RT-PCR	Cell based activity of SR-717: ISG-THP1, EC_50_ = 2.1 μM; ISG-THP1 cGAS KO, EC_50_ = 2.2 μM; ISG-THP1 STING KO, no activity up to the limit of solubility. SR-717 binds to STING with an apparent affinity IC_50_ = 7.8 μM. 30 mg/kg intraperitoneal once-per-day regimen of SR-717 for 1 week maximally inhibited tumour growth and prolonged survival in B16F10 model. The compound increased CD8+ T cells among TILs and in dLNs, as well as activated NK cells in dLN. SR-717 induced PD-L1 expression in THP1 cells and in primary human PBMCs. SR-717 STING agonist was found to induce IDO1 expression in primary human PBMCs.	[94]

* One unit of activity equals the amount of IFN in 1 mL of sample that reduced the viral lysis by 50% in the bioassay. Abbreviations: DMXAA, 5,6-dimethylxanthenone-4-acetic acid; FAA, flavone acetic acid; CMA, carboxymethyl-9-acridanone; BNBC, 6-bromo-*N*-(naphthalen-1-yl)benzo[*d*][1,3]dioxole-5-carboxamide; DSDP, 2,7,2″,2″-dispiro[indene-1″,3″-dione]-tetrahydro dithiazolo [3,2-a:3′,2′-d]pyrazine-5,10(5a*H*,10a*H*)-dione; diABZI, di-amidobenzimidazole; STING, stimulator of interferon genes; hSTING, human STING; mSTING, mouse STING; ITC, isothermal titration calorimetry; ELISA, enzyme-linked immunosorbent assay; q-RT-PPCR, real-time quantitative reverse transcription PCR; IFN, interferon; cGAMP, cyclic Gp(2′,5′)Ap(3′,5′); Kd, dissociation constant; PBMCs, peripheral blood mononuclear cells; EC_50_, half maximal effective concentration; KO, knock out; dLN, draining lymph node.

**Table 2 jcm-09-03323-t002:** STING Agonists in Clinical Trials.

Drug	Company	Cancer Type	Phase	Trial Start Date	Status (Estimated Completion)	Pertinent Findings of Trial	NCT Code
**ADU-S100 (i.t.) +/− ipilimumab (i.v.)**	Aduro Biotech; Novartis	Advanced/metastatic solid tumours; lymphomas	I	04/16	Terminated 12/19	Undisclosed	NCT02675439
**ADU-S100 (i.t.) + PDR001(i.v.) (spartalizumab)**	Novartis	Solid tumours; lymphomas	Ib	09/17	Terminated 12/19	Data cut-off: 5th April 2019 - 12/53 SD, 4/53 PR, 1/53 CR - Responders: median reduction of 73% in 1° lesion diameter - 78% TRAEs, 12.2% of TRAEs = grade3/4 - No DLTs - MTD not determined - T1/2 = 10–23 min	NCT03172936
**ADU-CL-20 (i.t.) + anti-PD-1 (i.v.)**	Aduro Biotech	Metastatic/recurrent HNSCC	II	08/19	Ongoing (2022)	Undisclosed	NCT03937141
**MK-1454 (i.t.) +/− pembrolizumab (i.v.)**	Merck & Co	Advanced/metastatic solid tumours; lymphomas	I	02/17	Ongoing (2021)	Data cut-off: 31st July 2018 - TRAEs 83% monotherapy, 82% combination - 7% in combination discontinued due to TRAEs - MTD not yet determined - Combination 6/25 (24%) → PR (3 HNSCC, 1 TNBC, 2 anaplastic thyroid carcinoma) - Combination: median reduction of 83% in 1° lesion diameter - T1/2 = 1.5 h	NCT03010176
**MK-2118 (i.t.; s.c.) +/− pembrolizumab (i.v.)**	Merck & Co	Advanced/metastatic solid tumours; lymphomas	I	09/17	Ongoing (2022)	Undisclosed	NCT03249792
**BMS-986301 (i.t.) +/− nivolumab (i.v.), ipilimumab (i.v.)**	Bristol-Myers Squibb	Advanced solid tumours	I	03/19	Ongoing (2023)	Undisclosed	NCT03956680
**GSK3745417 (i.v.; s.c.) +/− pembrolizumab (i.v.)**	GSK	Advanced solid tumours	I	03/19	Ongoing (2024)	Undisclosed	NCT03843359
**SB-11285 (i.v.) + nivolumab (i.v.)**	Spring Bank Pharmaceuticals	Advanced solid tumours	Ia/Ib	09/19	Ongoing (2022)	Undisclosed	NCT04096638
**IMSA-101 (i.t.) +/− ICI (i.v.)**	ImmuneSensor Therapeutics	Advanced solid tumours	I/IIa	09/19	Ongoing (2023)	Undisclosed	NCT04020185
**E7766 (i.t.)**	Eisai Inc.	Advanced solid tumours; lymphomas	Ia/Ib	03/20	Ongoing (2022)	Undisclosed	NCT04144140

Note: +/−, combination/alone. The NCT code refers to a unique identification code given to each clinical study registered on ClinicalTrials.gov. Abbreviations: GSK, GlaxoSmithKline; HNSCC; head and neck small cell carcinoma; SD, stable disease; PR, partial response; CR, complete response; TRAE, treatment related adverse event; DLT, dose limiting toxicity; ICI, Immune checkpoint inhibitors; MTD, maximum tolerated dose; TNBC, triple negative breast cancer.

**Table 3 jcm-09-03323-t003:** STING agonist drug delivery systems for cancer immunotherapies.

	Drug Delivery System	Loaded CDN	Tumour Models	ROA	Date	References
	YSK05 (pH sensitive cationic lipid with high fusogenicity)	c-di-GMP	B16-F10 (melanoma)	i.v.	08/15	[128]
**Liposomes**	YSK05 (pH sensitive cationic lipid with high fusogenicity)	c-di-GMP	E.G7-OVA (T cell lymphoma)	s.c.	04/14	[129]
	PEGylated lipid nanoparticles	c-di-GMP	EG.7-OVA (T cell lymphoma); B16-F10 (melanoma)	s.c.	05/15	[130]
	PEGylated cationic liposomes	2’3’-cGAMP	B16-F10 (melanoma)	i.v.; i.t.	01/17	[131]
	Soy-PC-DOTAP liposome	3’3’-cGAMP	C3(1) Tag model (basal-like TNBC); B16F10 (melanoma); C3(1) Tag GEM (basal-like TNBC)	i.v.	11/2018	[124]
	poly (beta-amino ester) (PBAE) *	ML-RR-S2-CDA (ADU-S100)	B16-F10 (melanoma)	i.t.	11/17	[123]
**Polymers**	PEG-DBP copolymers *	2’3’-cGAMP	B16-F10 (melanoma)	i.v.; i.t.	1/19	[14]
	PEG-DBP copolymers *	2’3’-cGAMP	Neuroblastoma	i.t.	03/20	[132]
	Ace-DEX microparticles	3’3’-cGAMP	E0771 (TNBC); B16-F10 (melanoma)	i.p.; i.m; i.v.; i.t.	06/19	[133]
	LPEI/HA	2’3’-cGAMP;3’3’-cGAMP	N/A	i.m.	10/15	[134]
**Hydrogels**	HA hydrogel scaffold	2’3’-cGAMP	4T1 (breast cancer)	i.v.; i.t.	03/18	[135]
	Matrigel	CDA	TC1 (lung cancer)	i.t.	11/18	[136]
	STINGel	ML-RR-S2-CDA (ADU-S100)	MOC2-E6E7 (Oral cancer)	i.t.	01/18	[137]

* Indicates the formulation was injected in combination with immune checkpoint inhibitor therapy as well. Abbreviations: CDN, cyclic dinucleotide; ROA, route of administration; s.c., subcutaneous; i.v., intravenous; i.t., intratumoral; i.p., intraperitoneal; i.m., intramuscular; TNBC, triple negative breast cancer; c-di-GMP, cyclic diguanylate; cGAMP, cyclic guanosine monophosphate–adenosine monophosphate; Soy-PC-DOTAP, (soy)L-α-phosphatidylcholine (Soy-PC) and 1,2-dioleoyl-3-trimethyl-ammonium-propane; Ace-DEX, Acetalated dextran; LPEI/HA, linear polyethyleneimine/hyaluronic acid; CDA, cyclic-di-adenosine 5′ monophosphate; PEG, poly(ethylene glycol); PEG-DBP, poly(ethylene glycol)-block-[(2-diethylaminoethyl methacrylate)-co-(butyl methacrylate)-co-(pyridyl disulphide ethyl methacrylate)].

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
