# Peer review of "Challenges and Opportunities in the Clinical Development of STING Agonists for Cancer Immunotherapy"

_jcm, 2020, doi:10.3390/jcm9103323_

Round 1

Reviewer 1 Report

This manuscript reviewed current progress in immunotherapy using STING agonist。 Clinical trials using STING agonist treating various types cancer have encountered some head winds lately and this review comes at the right time to discuss the challengers and opportunities in this research field. A comprehensive list of STING agonist is discussed in good level of details in the review with chemical structure, potential pharmacokinetics and clinical trial informations provided. Although many of those trials are still in early stage, the negative information broke out early this year indicated that substantial challenges may exist for all these agonists. Authors discussed potential used ICI combination therapy and using drug delivery systems to improve therapy. 

In general the the review is in very good quality. Below are a few suggestions for further improvement. 

  1. List full name for STING and cGAS.
  2. Line 50-52: IC is part of immune suppressive mechanisms in TME. The sentence stating ICI ineffetive in those type TME is inappropriate. Please rephrase
  3. Line 75: 2‘,3’-cyclic GMP-AMP should include full linkage information cyclic [G(2’,5’)pA(3’,5’)p]. cGAMP as abbreviation is fine afterwards
  4. Line 76: The first work "synthase" should be deleted. 
  5. In addition to IRF3, NKkB activation by STING should also be mentioned.
  6. Line 193-194, a reference is needed for trial involving MAVU-104
  7. Section 4.1, DMXAA:  DMXAA was not discovered as a sting agonist, rather a neo-vasculature disruption agent. Wether the anti-angiogenic effects are STING dependent is not clear and the reason for falure in phase III trail may be more than inability to bind hSTING. Please adjust the section accordingly. 
  8. Section 4.2, line 209: Early study on anti-neoplasitic function of CDN most likely is based on direct effects of IFN1 to suppress cells division or inducing cell death, which as very different from the current concept in inducing immune response from cancer.  This point need to be further clarified. 
  9. Line 215-216:  A reference need to be inserted for STING agonist eradicate malignant B cells 
  10. Line 262: "Mice bearing CT26 tumours showed no protection to tumour re-challenge" is a false statement. IN the Figure 5B of the cited study, CDA cured mice bearing CT26 also resist to re-challenge of CT26, but not a different tumour indicated immune memeory is tumour specific. Please correct
  11. Line 303-304. The strong therapeutic effect from diABZI in treating CT26 was show under the experimental condition that the drug is injected at merely 2 days after tumour engraftment when no tumour and TMEs were formed yet. The conclusion "system anti-tumour" effect need to cited with caution and providing some level of details for the clarification.   
  12. Line 351: "ICI-therapy has been shown to be ineffective in STING deficient mice" is also a incorrect statement.  ICI showed weaker therapy effects in STING deficient mice, not entirely infective (figure 6G in reference 12)
  13. Line 570: TC1 is mouse lung epithelial cells transformed by HPV16  E6/7, not a HNSCC model  please correct.
  14. A recently publication by Lemos et al (Overcoming resistance to STING agonist therapy to incite durable protective anti-tumor immunity) should be included as citation in the therapy resistance section.  

Reviewer 2 Report

This is an interesting review. The review discussed different immunotherapies targeting angle, it provides great background of immunotherapies. The review also discussed the challenge of clinical development of STING for cancer immunotherapy due to the nature of this against. Its a comprehensive review.

Only minor English editing is required. 

Reviewer 3 Report

The manuscript entitled “Challenges and opportunities in the clinical development of STING agonists for cancer immunotherapy" covers the recent studies on the development of STING agonists as a novel anti-cancer therapeutic and the current status of clinical trials using STING agonists. This review article is well-written and organized. The only minor comment is that the authors need to add the recent publication regarding orally available non-nucleotide STING agonists discovered by Merck and consider oral delivery for the future directions.   
